**Data Availability Statement:** All relevant data are within the paper and its Supporting Information files.

# Parasitic contamination of fresh vegetables and fruits sold in open-air markets in peri-urban areas of Jimma City, Oromia, Ethiopia: A community-based cross-sectional study

**Ahmed Zeynudin**[1], **Teshome Degefa**[1], **Tariku Belay**[1], **Jiru Batu Mumicha**[2], **Abdusemed Husen**[3], **Jafer Yasin**[4], **Abdulhakim Abamecha**[1]*, **Andreas Wieser**[5,6,7]

1 School of Medical Laboratory Sciences, Institute of Health, Jimma University, Jimma, Ethiopia, 2 Gumay Woreda Health Office, Jimma, Ethiopia, 3 Department of Oncology, Institute of Health, Jimma University, Jimma, Ethiopia, 4 Oda Hulle Primary Hospital, Jimma, Ethiopia, 5 Division of Infectious Diseases and Tropical Medicine, University Hospital, Ludwig-Maximilians-Universitat (LMU) Munich, Munich, Germany, 6 Department of Bacteriology, Max von Pettenkofer-Institute (LMU), Munich, Germany, 7 German Center for Infection Research (DZIF), Munich, Germany

* abdulhakimabamecha@gmail.com

## Abstract

### Background

Consuming contaminated raw vegetables and fruits is one of the primary means of parasite transmission to humans. Periodic monitoring of parasitic contamination in these food items is a crucial step in preventing the spread of parasitic disease in the community. This study was aimed at detecting intestinal parasitic contamination caused by consuming raw vegetables and fruits sold in three open-air markets and its associated factors in peri-urban areas of Jimma City, Oromia, Ethiopia.

### Methods

A cross-sectional study was conducted on fruits and vegetables collected from three peri-urban open-aired markets (namely; Hora Gibe, Bore and Jiren markets) in peri-urban areas of Jimma City between July and September 2021. A total of 187 fresh vegetable samples and 188 fruits were collected and examined for intestinal parasite contamination. About 200g of fruit and vegetable samples were processed and examined microscopically for parasite contamination, utilizing direct wet mount and modified Zeihl-Neelson staining methods in accordance with standard protocols. A structured questionnaire was used to collect data on the socio-demographic characteristics of vendors and risk factors for fruit and vegetable contamination. All data were analyzed using SPSS version 20.0.

**Funding:** The authors received no specific funding for this work.

**Competing interests:** The authors have declared that no competing interests exist.

## Result

Of the 187 fresh vegetable samples and 188 fresh vegetable samples, 105 (56.1%) and 68/188 (36.2%) of vegetables and fruit samples, respectively, were found contaminated with one or more intestinal parasites. Remarkably, high level of contamination in fresh vegetable samples was recorded both in Carrot (*Daucus carota*) 63.8% (30/46) and Lettuce (*Lactuca sativa*) 63.1% (29/46) while Green pepper (*Capsicum* spp.) is the least contaminated. In fruit samples, Avokado (*Persea americana*) 42.6% (20/47) and Banana (*Musa acuminata*) 14.9% (7/47) were the most and the least commonly contaminated items respectively. The identified helminthes and protozoans were *Ascaris lumbricoides*, *Strongyloides stercoralis*, *Hymenolepis nana*, *Entamoeba histolytica/dispar*, *Giardia lamblia*, *Cryptosporidium* spp., *Toxocara* spp. And *Fasciola* spp. The most predominant parasite encountered was *A. lumbricoides 46(12.3%)* whereas *both Toxocara* spp. 12(6.9) and *Fasciola* spp. 2(0.5) were the least detected parasites. It is worth-mentioned that the rate of contamination in Bore market (38.15%) was higher compared with Jiren market (34.7%) and Hora Gibe market (27%). However, the rate of contamination in vegetables and fruit obtained from the three district was non-significant ($p$ = 0.19). Contamination was more common in vegetables than fruits (AOR = 5.78, p<0.001). It was also observed that decreased parasitic contamination was significantly associated with washing the products before displaying it for selling ($p$ < 0.001).

## Conclusion

The study has identified a high rate of raw vegetables and fruits contaminated with intestinal helminthes and protozoan. Contaminated fresh vegetables and fruits in open-aired peri-urban markets of Jimma city, Ethiopia may play a significant role in transmission of intestinal parasitic infections to humans, particularly *A. lumbricoides* infection. Therefore, it is urgently needed for health authorities to educate the public on the proper handling of vegetables and fruits prior to consumption.

## Background

Intestinal parasitic infections are one of the most significant public health problems globally, affecting approximately 3.5 billion people and causing over 450 million illnesses annually [1,2]. Most diseases caused by the intestinal parasites (helminths and protozoa) have been categorized as Neglected Tropical Diseases (NTDs) that have been a significant public health problem in many developing countries, including Ethiopia [1–4].

Intestinal parasites (Ips) are primarily transmitted by fecal-oral routes, mostly via ingestion with contaminated food and water or during direct hand-to-mouth contact [5,6]. It has been reported that food-borne parasitic infections are associated with the consumption of contaminated fresh vegetables [6]. In general, vegetables and fruits are considered to be vehicles that easily transmit parasites into individuals, especially when eaten raw or without peeling [7]. Studies conducted on various items of fruit and vegetable samples have shown that *Ascaris lumbricoides*, *Cryptosporidium* spp., *Entamoeba histolytica*, *Enterobius vermicularis*, *Fasciola* spp., *Giardia lamblia*, hookworms, *Hymenolepis* spp., *Taenia* spp., *Trichuris trichiura*, *Cyclospora* spp., and *Toxocara* spp. Infect humans who consume contaminated fruits and vegetables without cooking or washing them properly [5]. This problem is becoming an increasing

concern because of the expanding number of susceptible people (i.e., the elderly and the immune-compromised), more extensive produce trade across international borders, and changes in national and international policies concerning food safety[5–7].

Vegetables and fruits become contaminated with various parasitic stages through three major pathways: contamination of raw vegetables and fruits on the farm during harvesting; contamination of water (use of human and animal excreta as natural fertilizer, and untreated waste water) used for irrigation or washing process; and contamination of food handlers/vendors [6–8]. Factors in the post-harvesting phase include storage, transportation, and marketing conditions as well as hygienic practice during processing for consumption in food service or home settings [6–8]. In developing countries such as Ethiopia, poor water supply, hygiene and sanitary practices, and sub-standard and crowded living conditions lead to an increased risk of acquiring parasitic infections [6]. Hence, it is expected that farmlands will be contaminated with infective Ips mainly due to open defecation. In addition, natural fertilizer (human and animal excreta) is commonly used by farmers in the country, and water used for irrigation is usually contaminated. Moreover, fruits and vegetables such as bananas, mangoes, tomatoes, salads, and green peppers are frequently consumed raw. All these contribute to parasitic contamination of fruits and vegetables so that these food items serve as important vehicles for transmission in humans [6].

There are plenty of open markets where fruits and vegetables are sold in Africa. Fresh produce is exposed to the environment in these markets, including domestic animals, which add to the risk of food contamination [6,9–11]. On this continent, studies on the prevalence of intestinal parasites associated with vegetables and fruits have been performed mostly in Mozambique, and Ghana [9–11]. A fragmented and limited number of studies have been conducted in Ethiopia to evaluate the parasitic contamination of vegetables and fruits [6]. The results of these studies showed inconsistency and variation in prevalence: 25.1% to 57.8% of the fruit and vegetable samples collected at the marketing phase were contaminated with Ips [6]. However, the rate of contamination and species of parasites vary by weather conditions, socio-cultural status, season of sample collection, fruit and vegetable items examined, and other factors. This demands monitoring of the contamination status and contributing factors at local settings in order to intervene in the transmission of Ips. Despite this, there is no adequate data in the peri-urban areas of Jimma city. Hence, the aim of the present study was to assess the parasitic contamination rate of vegetables and fruits and associated factors in purposively selected local markets in peri-urban areas of Jimma city, in an effort to help authorities find out the contamination level and to design comprehensive monitoring and educational programs according to the need.

## Method

### Study design, area and period

A cross-sectional study was conducted in three peri-urban areas (Hora Gibe, Bore, and Jiren markets) of Jimma city from July to September 2021(during the major rainy season). Jimma city is about 345 kilometers away from the capital city, Addis Ababa, in the southwest direction. The town is located at 7º 40' North latitude and 36º 5' East Longitude, and the climate condition is relatively cool tropical monsoon climate with an average altitude of about 1780 m above sea level, a mean annual maximum temperature of 30˚C and a mean annual minimum temperature of 14˚C. The annual rainfall ranges from 1138 mm to 1690 mm. The fresh vegetables and fruits sold in these markets were brought from different areas of Jimma Zone agricultural areas.

## Sample collection

A total of 375 fresh vegetable and fruit samples, including eight different types that are frequently consumed in the area were randomly purchased from vendors in three purposively selected open-air markets. The fresh raw vegetable and fruit samples used in this study included Lettuce (*Lactuca serriola*), Cabbage (*Brassica oleracea*), Carrot (*Daucus carota*), Avocado (*Persea Americana*), Tomato (*Lycopersicon esculentum*), Green pepper (*Capsicum annuum*), Banana (*Musa paradisiaca*) and Mango (*Mangifera indica*). An equal number of samples (almost 47 each, a total 373 samples) were collected from three purposively selected open-air markets. Each sample was placed in a sterile polythene bag, properly labeled, and immediately transported to the Medical Microbiology and Medical Parasitology Laboratory of Jimma University for parasitological examination. A pre-tested structured questionnaire was used to collect data on predisposing factors for parasitic contamination of vegetables and fruits at local markets.

## Parasitological examination

To detect the presence of parasites in the studied samples, we followed a standard protocol explained elsewhere [6,12,13]. In brief, about 200g of each vegetable and fruit were washed separately in 500mL of normal saline to detach the parasitic stages (ova, larvae, cysts, and oocysts) of helminths and protozoan parasites commonly assumed to be associated with vegetable contamination. After overnight sedimentation of the washing solution, 15 mL of the sediment was transferred to a centrifuge tube using a sieve, to remove undesirable matters [6]. The tube was centrifuged at 3000 rpm for five minutes to concentrate the parasitic stages. [6]. Then, the supernatant was decanted carefully without shaking, and the sediment was agitated gently by hand to re-suspend the sediment. Part of the sediment was used for direct and iodine wet mount smear preparation and examined under a light microscope using 10X and 40X objectives for the detection and identification of parasitic ova and larvae, using coloured parasitological atlases as guide. The remaining sediment was processed and examined by the modified Ziehl–Neelsen staining technique for the detection of coccidian and *Cryptosporidium* oocysts following the standard protocol explained elsewhere [6,12,13]. Two slides were prepared and examined per sample in both the direct wet mount and modified Ziehl–Neelsen staining techniques.

## Data analysis

Data was entered in Epi Info version 3.5.3 and exported to SPSS version 20. Descriptive statistics like frequency and proportion were calculated to explain the characteristics of vendors and the contamination status of vegetables and fruits. Binary logistic regression analysis was done to assess factors associated with fruit and vegetable contamination. Variables with a *p*-value <0.25 in the binary logistic regression analysis were taken as candidates for multiple logistic regression in order to avoid the effect of confounders. An association with a p-value <0.05 at 95%CI was considered as significant.

## Ethical considerations

Prior to the study implementation, ethical clearance was obtained from Institutional Ethics Board (IRB) of the Institute of Health, Jimma University, (Ref No; IHRPGn/357/2021). Participation was voluntary. The Ethics Committee indicated no need to obtain a written informed consent in this survey. The questionnaire was anonymous; therefore, any document as a written informed consent that might reveal the identity of the subjects was asked. However,

subjects provided a verbal consent obtained from the sellers/vendors to participate in the study, and people accompanying them, including relatives, are witnessed.

## Results

A total of 187 fresh vegetables and 188 fruit samples were examined for the presence of parasite contamination. The prevalence of parasitic contamination in different vegetable and fruit samples among the three markets is shown in **Table 1** and S1 Fig. Helminthic eggs and protozoan cysts were detected in 56.1% (105/187) of fresh vegetable samples and in 36.2% (68/188) of fruit samples examined; the overall contamination rate in all vegetable and fruit samples was 46.1%.

The parasites were detected in 67.3%, 73.7%, and 37% of vegetable samples obtained from Hora Gibe, Bore, and Jiren markets, respectively. The highest rate of parasitic contamination in vegetables was found in the Bore market, whereas the lowest rate was observed in the Jiren market. However, the contamination rate of samples collected from the three markets was statistically not significant ($p = 0.053$).

The parasites detected include ova of *Ascaris lumbricoides*, *Strongyloides stercolaris*, *Hymenolepis nana*, *Toxocara* spp., *Fasciola* spp.*; cysts of G. intestinalis* and *E. histolytica/dispar*; and oocysts of *Cryptosporidium* spp. **Table 2** shows that *Ascaris lumbricoides* (23.1%) was the most frequently detected parasite, followed by *S. stercoralis larvae* (18.5%), *E.histolytica/dispar cyst* (15%), *Giardia* spp**.** (14.5%), *Hymenolepis nana* (11.6%), *Cryptosporidium* (9.2%) oocysts, *Toxocara* spp. egg (6.9%), and *Fasciola* spp. egg (1.1%).

Table 3 shows the distribution of intestinal parasites in relation to the type of fresh vegetable and fruit samples collected from the three open-air markets. The highest prevalence of parasites was detected in carrot 63.8% (30), followed by lettuce 63.1% (29) and Cabbage 61.7% (29). Banana with a prevalence of 14.9% (7) was the least contaminated fruit. Among the eight vegetable and fruit types included in the study, *A. lumbricodes* was abundantly detected in samples of tomato 16/16 (100%), followed by cabbage 11/24(45.8%) and lettuce 5/24(20.8%).

In the present study, interviews of the vendors have been conducted to evaluate the association of parasitic contamination of vegetables and fruits in the markets (See **S1 Table**). Vendors were asked about their educational status and it was revealed that 14.1% of the vendors had no

**Table 1. Distribution of intestinal parasitic contamination in different fresh vegetables and fruits among the three open aired markets in peri-urban areas of Jimma city from July-September, 2021.**

| Product type | No. of examined samples | | Hora Gibe Market | | Bore Market | | Jiren Market | |
|---|---|---|---|---|---|---|---|---|
| | Examined | Positive (%) | No. of examined | No. of positives (%) | No. of examined | No. of positives (%) | No. of examined | No. of positives (%) |
| **Vegetables** | | | | | | | | |
| Lettuce | 46 | 29(63.0) | 16 | 14(87.5) | 11 | 9(81.8) | 19 | 6(31.6) |
| Cabbage | 47 | 29(61.7) | 17 | 12(70.6) | 10 | 7(70.0) | 20 | 10(50.0) |
| Carrot | 47 | 30(63.8) | 2 | 1(50.0) | 22 | 19(86.4) | 23 | 10(43.5) |
| Green pepper | 47 | 17(36.2) | 14 | 6(42.8) | 14 | 7(50.0) | 19 | 4(21.1) |
| **Total** | **187** | **105(56.1)** | **49** | **33(67.3)** | **57** | **42(73.7)** | **81** | **30(37.0)** |
| **Fruits** | | | | | | | | |
| Tomato | 47 | 23(48.9) | 9 | 0(0.0) | 30 | 22(73.3) | 8 | 1(12.5) |
| Banana | 47 | 7(14.9) | 10 | 1(10.0) | 26 | 0(0.0) | 11 | 6(54.5) |
| Mango | 47 | 18(38.3) | 17 | 4(23.5) | 14 | 1(7.1) | 16 | 13(81.3) |
| Avocado | 47 | 20(42.6) | 15 | 8(53.3) | 16 | 1(6.3) | 16 | 11(68.8) |
| **Total** | **188** | **68(36.2)** | **51** | **13(25.5)** | **86** | **24(27.9)** | **51** | **31(60.8)** |
| **Overall total** | **375** | **173(46.1)** | **100** | **46(46.0)** | **143** | **66(46.2)** | **132** | **61(46.2)** |

**Table 2. Distribution of intestinal parasites contamination in fresh vegetables and fruits samples collected from open aired markets in peri-urban areas of Jimma city from July-September, 2021.**

| Detected parasites | Total prevalence in vegetables (n = 105) | Total prevalence in fruits(n = 68) | Total prevalence in both vegetables and fruits (N = 173) |
|---|---|---|---|
| | n (%) | n (%) | n (%) |
| *A. lumbricodes* | 24(22.9) | 16(23.5) | 40(23.1) |
| *S. stercoralis* | 18(17.1) | 14(20.6) | 32(18.5) |
| *E. histolytica/dispar* | 17(16.2) | 9(13.2) | 26(15.0) |
| *Giardia* spp. | 16(15.2) | 9(13.2) | 25(14.5) |
| *H. nana* | 10(9.5) | 10(14.7) | 20(11.6) |
| *Cryptosporidium* spp. | 9(8.6) | 7(10.3) | 16(9.2) |
| *Toxocara* spp. | 9(8.6) | 3(4.4) | 12(6.9) |
| *Fasciola* spp. | 2(1.9) | 0(0.0) | 2(1.2) |

formal education, 25.9% had primary education, and 6.1% had secondary and above education. No significant association was noted between the education level of vendors and the parasitic contamination rate of the vegetables they were selling (see Table 4). Among the factors associated with parasitic contamination of vegetables and fruits is the act of washing products before displaying it for sale. This study showed that the majority (44.0%) of the products were not washed before display, with only 2.0% being washed. By putting all other variables constant, the likelihood of parasitic contamination of vegetables and fruits is 251.8 times in vegetables and fruits not washed than washed before display (95% C.I. 53.4–1187.4, *p*-value = <0.001), 5.8 times in vegetables than fruits (C.I. 1.6–20.2, *p*-value = 0.006), 54.8 times in vegetables and fruits displayed on the floor than on the table/shelf (C.I. 10.1–296.7, *p*-value = <0.001), 21.7 times in vegetables than fruits collected from venders who had no formal education than who had secondary and above education (C.I. 1.8–28.2, *p*-value = 0.018) (see Table 4).

## Discussion

The consumption of raw vegetables and fruits plays an important role in the transmission of parasites to humans, especially in countries that frequently lack good sanitization and personal

**Table 3. Distribution of intestinal parasites in relation to the type of fresh vegetable and fruit samples collected from the three open aired markets in peri-urban areas of Jimma city from July-September, 2021.**

| Product type | Helminthes' eggs and larvae | | | | | Protozoan oo(cysts) | | | Prevalence (%) |
|---|---|---|---|---|---|---|---|---|---|
| | *A. lumbricodes* | *S. stercoralis* | *H. nana* | *Toxocara* spp. | *Fasciola* spp. | *E. histolytica/dispar* | *Giardia* spp. | *Cryptosporidium* spp. | |
| **Vegetables** | | | | | | | | | |
| Lettuce | 8 | 8 | 3 | 3 | 0 | 3 | 3 | 1 | 29(63.1) |
| Cabbage | 11 | 4 | 0 | 14 | 0 | 0 | 0 | 0 | 29(61.7) |
| Carrot | 5 | 3 | 7 | 0 | 0 | 0 | 3 | 12 | 30(63.8) |
| Green pepper | 0 | 3 | 0 | 0 | 2 | 6 | 3 | 3 | 17(36.1) |
| Total | 24 | 18 | 10 | 17 | 2 | 9 | 9 | 16 | 105(56.1) |
| **Fruits** | | | | | | | | | |
| Tomato | 16 | 0 | 0 | 0 | 0 | 7 | 0 | 0 | 23(48.9) |
| Banana | 0 | 3 | 0 | 3 | 0 | 0 | 0 | 1 | 7(14.9) |
| Mango | 0 | 0 | 9 | 3 | 0 | 0 | 3 | 3 | 18(38.3) |
| Avocado | 0 | 11 | 1 | 3 | 0 | 0 | 0 | 5 | 20(42.6) |
| Total | 16 | 14 | 10 | 9 | 0 | 7 | 3 | 9 | 68(36.2) |
| **Overall total** | **40** | **32** | **20** | **26** | **2** | **16** | **12** | **25** | **173(46.1)** |

**Table 4. Binary and multiple logistic regressions of factors affecting parasitic contamination of vegetables and fruits samples collected from the three open aired markets in peri-urban areas of Jimma city from July-September, 2021.**

| Variable | Category | Parasitic contamination | | COR(95% CI) | p-value | AOR (95%CI) | p-value |
|---|---|---|---|---|---|---|---|
| | | Yes Freq (%) | No Freq (%) | | | | |
| Educational status of vendors | No formal education | 53(14.1) | 5(1.3) | 52.1(18.8–144.5) | <0.001* | 21.7(1.8–28.2) | 0.02* |
| | Primary | 97(25.9) | 84(22.4) | 5.8(1.1–33.2) | <0.001* | 1.5(0.4–5.7) | 0.57 |
| | Secondary and above | 23(6.1) | 113(30.1) | 1 | | 1 | |
| Site collection | Bore | 66(46.2) | 143(11.5) | 1 | | | |
| | Hora Gibe | 46(46.0) | 100 (26.7) | 1.0(0.6–1.6) | 0.90 | | |
| | Jiren | 61(46.2) | 132(35.2) | 1.1(0.6–1.8) | 0.82 | | |
| Source of vegetables and fruits | Farmers | 113(30.1) | 99(26.4) | 1.1(0.3–4.1) | 0.84 | | |
| | Middle men | 55(14.7) | 98(26.1) | 0.6(0.6–2.0) | 0.38 | | |
| | Private garden | 5(1.3) | 5(1.3) | 1 | | | |
| Hygienic condition of sellers | Adequate | 19(5.1) | 99(26.5) | 1 | | 1 | |
| | Not adequate | 154(41.1) | 103(27.5) | 7.8(4.5–13.5) | <0.001* | 2.5(0.6–11.2) | 0.22 |
| Wash status | Yes | 8(2.1) | 152(40.5) | 1 | | 1 | |
| | No | 165(44.0) | 50(13.3) | 62.7(28.8–135.5) | <0.001* | 251.8(53.4–1187.4) | <0.00* |
| Type of Products | Vegetables | 105(28) | 82(21.9) | 2.9(1.9–4.6) | <0.001* | 5.8(1.6–20.2) | 0.01* |
| | Fruits | 68(18.1) | 120(32) | 1 | | 1 | |
| Means display | Floor | 167(44.5) | 121(32.1) | 18.6 (7.9–44.1) | <0.001* | 54.8(10.1–296.7) | <0.00* |
| | Table/Shelf | 6(1.6) | 81(21.6) | 1 | | 1 | |
| Knowledge | Good Knowledge | 111(29.6) | 54(14.4) | 1 | | 1 | |
| | Poor Knowledge | 62(16.5) | 148(39.5) | 4.9 (3.2–7.6) | <0.001* | 0.3 (0.1–1.2) | 0.09 |
| Attitude | Positive attitude | 57(15.2) | 138(36.8) | 1 | | 1.1(0.3–3.8) | 0.86 |
| | Negative attitude | 116(30.9) | 64(17.1) | 4.4 (2.8–6.8) | <0.001* | 1 | |
| Practice | Good practice | 17(4.5) | 135(36.0) | 1 | | 1 | |
| | Poor practice | 156(41.6) | 67(17.9) | 18.5(10.4–33.0) | <0.001* | 6.3(1.8–22.8) | 0.01* |

hygiene practices. The recovery of parasites from vegetables and fruits helped us better understand the potential source of intestinal parasitic acquisition among a community. In the present work, four types of raw vegetables: lettuce (*Lactuca serriola*), cabbage (*Brassica oleracea*), carrot (*Daucus carota*), green pepper (*Capsicum* spp.) and four types of fruits: avocado (*Persea Americana*), tomato (*Lycopersicon esculentum*), banana (*Musa paradisiaca*) and mango (*Mangifera indica*) that are commonly sold and consumed among the population in Hora Gibe, Bore, and Jiren markets were examined.

This study demonstrated that the overall prevalence of parasitic contamination in vegetables and fruits sold in three markets was 46.1%. This rate is approximately similar to the previous study conducted in Bahidar City, Northwest Ethiopia (39.1%) [6], and consistent with previous findings from Saudi Arabia (46%) [14] and Iraq (51%) [15]. However, as compared to a similar study conducted in Kenya, the present study showed a low contamination rate (46.1% vs. 75.9%) [16]. Our finding is explained by the fact that such markets have more frequent running water during the main rainy season, which increases the likelihood that producers and vendors will wash their products thoroughly. Another hypothesis is assuming that parasite forms on the surface of vegetables are washed away by rain [9]. Moreover, since 2015, water, sanitation, and hygiene (WASH) activities, health education, and biannual deworming programs for helminthes have been conducted and substantially decreased environmental

fecal contamination as well as parasitic contamination in Ethiopia [6]. The difference may also be due to variations in the types and quantities of fruits and vegetables evaluated.

Results from this study show that the Bore market recorded the highest prevalence (73.7%) of intestinal parasites in fresh vegetable samples, followed by Hora Gibe market (67.3%) and Jiren market (37.0%). Nevertheless, there was no statistically significant association between the location of sale and the rate of contamination. Of course, since using night soil is not common in this area, these vegetables are cultivated in the gardens and fertilized with non-treated night soil. The differences between the three markets might be due to the different sources of vegetables as well as the hygienic practices in handling and washing by different sellers. Moreover, the high prevalence of intestinal parasites on vegetables is due to the fact that these open-air markets were characterized by poor sewage disposal systems, and vegetables were brought in from rural areas around these markets.

Of the eight vegetable and fruit types included in the study, the highest prevalence of intestinal parasites was detected in carrot 63.8%, followed by lettuce 63.1% and cabbage 61.7%, were the most contaminated with various parasites. Banana was the least contaminated fruit (14.9%). Variation in the occurrence of parasitic contamination among the different vegetable and fruit samples examined in this study could arise from differences in plant shape and surface area. Vegetables such as carrot, cabbage, and lettuce have a larger and uneven surface area that allows parasitic eggs and cysts to easily attach to the vegetable. On the other hand, vegetables like Green pepper with long narrow leaves and fruits with smooth leathery surfaces like Banana have the lowest occurrence of parasitic contamination because their narrow or smooth surfaces reduce the rate of parasitic attachment [17].

In this present study, *Ascaris lumbricoides* was detected in 23.1% (40/173) of all vegetables and fruits examined and were the most predominant intestinal parasite. The rate of contamination with *A. lumbricoides* were 68% in Tripoli, Libya [18], 20.83% in Arba Minch town, southern Ethiopia [17], 8.17% in Shahrekord, Iran [19] and 1.3% in Bahir Dar City, Northwest Ethiopia [6] although the incidence of ascariasis has decreased in recent year, and the national deworming program against soil-transmitted helminthes might bring this shift [17]. However, vegetable and fruit contamination in this study may indicate the potential risk of its reemergence in the study area. Contamination in vegetable and fruit samples might occur at any point along the chain; during planting, harvesting, transportation, or the marketing of vegetables. *Ascaris ova* are very resistant to environmental factors and are transmitted to consumers easily [17].

The second most prevalent contamination found in this study was the larvae of *S. stercoralis* (18.5%). Our finding was consistent with previous studies in Jimma city, Ethiopia, which reported a 21.9% prevalence of *S. stercoralis* contamination in vegetables and fruits [20]. A high rate of *Strongyloides* spp. Contamination might be because Strongyloides spp. Has a complex life cycle with a free-living stage in the environment that does not require a host for its proliferation [21].

*Entamoeba histolytica/dispar cysts* (15%) was the third frequently detected parasite followed by *Giardia* spp**.** (14.5%) which is consistent with previous finding from Bahir Dar City, Ethiopia [6]. However, the rate of contamination was lower as compared to a study finding from Arba Minch, Ethiopia [13]. Variation in the prevalence of *E. histolytica/E. dispar* and *Giardia* spp. Might be attributed to the long periods of survival of the cysts under cool and moist conditions and variations in geographical distribution [6].

*Hymenolepis nana* ranked as the fourth frequently detected parasite in the present study, with a prevalence of 11.6%, which was comparable to the prevalence of 11.9% and 15.56% from previous reports in Tarcha town [22] and in Arba Minch town [17], in Ethiopia,

respectively. However, in contrast with previous study conducted in Jimma (8.3%) [20], climatic variations and geographic difference might have contributed for the discrepancy observed [6,22].

The occurrence of *Cryptosporidium* oocysts in the different vegetable and fruit samples examined was 9.2%, which is in contrast with the rate of contamination reported from Ghana (17%) [23] and Ethiopia (12.8%) [20]. On the other hand, it is lower when compared with the finding from Arba Minch town, southern Ethiopia (5.8%) [13]. The discrepancy between the present study and previous studies might be as a result of the variations in geographical locations, climatic and environmental conditions, the kind of sample and sample size examined, the sampling techniques, methods used for detection of the intestinal parasites, and socioeconomic status. So long as these factors differ, consequently the discrepancy of the results would be expected [20,24]. The protozoan parasite *Cryptosporidium* is widespread, and water plays a crucial role in the spread of this intestinal infection [24]. The surface water is more likely to be contaminated with human and/or animal feces in the environment. It can also be contaminated by feces entering the agricultural run-off of adjacent farm animals or from human sewage. Contact of the vegetables and fruits with soil may also play a significant role in the contamination of the vegetables with *Cryptosporidium* oocysts as the vegetables were kept in a dirty environment and in contact with soil. Furthermore, the reinforcement of horticultural crops, such as vegetables, with cattle and sheep dung (dried fecal matters) containing viable *Cryptosporidium* oocysts posed a significant risk of vegetable contamination [24].

This study also detected *Toxocara* spp. Eggs in 6.9% (12/173) of the fresh vegetable and fruit samples examined. Human toxocariasis is a helminthic zoonotic disease caused by larval stages of *Toxocara canis* and less frequently by *Toxocara cati* [25,26]. The long-term survival of *Toxocara* eggs outside their hosts, coupled with their high fecundity, is responsible for significant contamination of soil with infective eggs [25,26]. This study indicates that pet animals (dogs and cats), which are the source of *Toxocara* eggs, may at some point shed contaminated feces onto farming areas.

The prevalence of *Fasciola* spp. Was 1.1% in this study, which was comparable to the prevalence of 1.8% from a similar study in Bahir Dar, Ethiopia [6]. A lower recovery rate of 0.3% was detected in similar studies in Ghana [27]. Fascioliasis is known to be a neglected zoonosis [27].

Evaluation of hygiene and handling practices of vendors was also assessed in this study. Accordingly, fruits and vegetables washed before display, type of products, means of display, educational status, and practice of sellers/vendors showed statistically significant associations with parasitic contamination (Table 4). We observed that vegetables were more contaminated with parasites than fruits. The higher prevalence of parasitic contamination in vegetables may be due to the rough surfaces and leaf folds of vegetables, which may retain dirt that cannot be easily washed off [6]. The results of this study indicate that there was good awareness (knowledge) and a positive attitude towards intestinal parasitic contamination, but the practice was poor, which is in agreement with other authors who report that post-harvest contamination can occur during transport and handling of the products [9]. The handling of products during preparation for display in the supermarket or even by the consumer can be one of the factors for the high level of contamination that we found [9].

Contamination with multiple species of medically important parasites was observed in all types of vegetables and fruits studied. This might point out the continued existence of intestinal parasitic infections in the area. Hence, the results of this study emphasized that raw vegetables and fruits, could be possible vehicles of parasitic transmission to humans. However, it is important to note that our study has several limitations. This study did not demonstrate the effect of seasonal variation on parasitic contamination; this factor might have affected the rate of

vegetable contamination in our study. Besides, no attempt was made to speciate the identified parasites by molecular methods, which is necessary to determine their public health threat.

## Conclusion

This study demonstrated that there is a high prevalence (46.1%) of parasitic helminth and protozoa contamination of fruits and vegetables in the study area, which could imply that, approximately, one out of every two vegetables or fruits was contaminated with helminthes and protozoa. This therefore highlights the significance of raw vegetables and fruits as potential transmission means for parasitic helminths and protozoa. This finding raises concern about public health being at high risk of infection, with neglected parasitic infections including ascariasis, strongyloidiasis, and others. This may be the result of direct human contamination, or contaminated soil and water harbored by the sampled vegetables and fruit during cultivation, transport, and handling/preparation by vendors. Given the public health and veterinary burden associated with both parasites, it is important to identify the factors that are crucial in reducing the spread of these parasites in communities and environments using one health approach, and it is also important to conduct awareness campaigns about parasite contamination and its various sources.

## Supporting information

**S1 Fig. Distribution of positive samples in fruits and vegetables among study sites.**
(DOCX)

**S1 Table. Socio-demographic and characteristics of the study participants in peri-urban areas of Jimma city from July-September, 2021.**
(DOCX)

## Acknowledgments

We would like to extend our gratitude to the vendors for their willingness and cooperation during the study.

## Author Contributions

**Data curation:** Ahmed Zeynudin, Teshome Degefa, Jiru Batu Mumicha.

**Formal analysis:** Ahmed Zeynudin, Teshome Degefa, Tariku Belay, Jiru Batu Mumicha, Abdusemed Husen, Jafer Yasin, Abdulhakim Abamecha.

**Investigation:** Ahmed Zeynudin, Teshome Degefa, Tariku Belay, Jiru Batu Mumicha, Abdusemed Husen, Jafer Yasin, Abdulhakim Abamecha, Andreas Wieser.

**Methodology:** Ahmed Zeynudin, Teshome Degefa, Tariku Belay, Jiru Batu Mumicha.

**Project administration:** Ahmed Zeynudin.

**Supervision:** Ahmed Zeynudin, Tariku Belay, Andreas Wieser.

**Writing – original draft:** Jiru Batu Mumicha.

**Writing – review & editing:** Ahmed Zeynudin, Teshome Degefa, Tariku Belay, Jiru Batu Mumicha, Jafer Yasin.

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
