## [Decision Letter · Decision Letter 0]

9 Mar 2023

PONE-D-23-01852Parasitic contamination of fresh vegetables and fruits sold in open-air markets in peri-urban areas of Jimma City, Oromia, Ethiopia: A community-based cross-sectional studyPLOS ONE

Dear Dr. Abamecha,

Thank you for submitting your manuscript to PLOS ONE. After careful consideration, we feel that it has merit but does not fully meet PLOS ONE’s publication criteria as it currently stands. Therefore, we invite you to submit a revised version of the manuscript that addresses the points raised during the review process.

We look forward to receiving your revised manuscript.

Kind regards,

Adriana Calderaro

Academic Editor

PLOS ONE

Journal Requirements:

2. In the ethics statement in the Methods, you have specified that verbal consent was obtained. Please provide additional details regarding how this consent was documented and witnessed, and state whether this was approved by the IRB

Reviewers' comments:

Reviewer's Responses to Questions

**Comments to the Author**

1. Is the manuscript technically sound, and do the data support the conclusions?

Reviewer #1: Yes

Reviewer #2: Yes

2. Has the statistical analysis been performed appropriately and rigorously? 

Reviewer #1: Yes

Reviewer #2: Yes

3. Have the authors made all data underlying the findings in their manuscript fully available?

Reviewer #1: Yes

Reviewer #2: Yes

4. Is the manuscript presented in an intelligible fashion and written in standard English?

Reviewer #1: Yes

Reviewer #2: Yes

5. Review Comments to the Author

Reviewer #1: This study has been well carried out and shows how much dirty vegetables and fruits could transmit parasitic infections to humans. The experimentally procedure and statistically analysis is done with classical methods and the English grammatical form is corrected and well understandable.

The only note to Authors is: how do you can distinguish Giardia cysts at specific level with only the microscopical assay? It could be better to indicate this parasite presence with Giardia sp. definition. Please, note also that Cryptosporidium is incorrectly written (Cyreptosporidium) and that spp. does not require capital letter (Spp), in both Figure 2 and 3.

Reviewer #2: Major comments:

All rounding to decimal places must be checked as there are errors in all tables.

The abbreviation “spp.” must always be written in lower case and not italics. Do the respective changes in all text and tables.

Authors should always consider using the same designation for Giardia, lamblia or intestinalis (only one name), throughout the entire text.

The authors do not explain whether the collection of samples was made in the rainy season or in the dry season, or summer or winter. The fact that they say that the collection was made from July to September does not indicate, for the reader, the degree of rainfall and temperature.

In the introduction, results and discussion, the authors did not take into account a similar study that was carried out in Mozambique and published in 2021, involving similar populations. This study should be introduced in this article and compared with the results you obtained because it is a recent study, carried out in East Africa..

“Salamandane et al.,. (2021). Occurrence of intestinal parasites of public health significance in fresh horticultural products sold in Maputo markets and supermarkets, Mozambique. Microorganisms, 9, 1806. https://doi.org/10.3390/microorganisms9091806”

Minor comments:

Background:

Line 105 – “vegetables and fruits….”

Parasitological examination:

Line 145, 147 – “modified Ziehl–Neelsen….”

Line 145 – Since Cryptosporidium is not a coccidia, authors should write “detection of coccidian and Cryptosporidium oocysts….”

Results:

Line 162 – “37.0%” instead of “37.3%”

Line 172 – “ Cryptosporidium spp.”

Lines 173-174 – “S. stercoralis”

Line 175 – “Cryptosporidium (9.2%) oocysts”

Line 175 – which forms of development were identified for Toxocara and Fasciola?

Lines 181-182 – “detected in carrot 63.8% (30), followed by lettuce 63.1% (29) and Cabbage 61.7% (29). Banana with a prevalence of 14.9% (7)…”

Line 183 – “A. lumbricodes was abundantly…”

Discussion:

Lines 211-214 – The names of vegetables and fruits should be in low case “lettuce….”

Line 221 – Authors should discuss the difference of contamination rate found in their study and in the study done also in Kenya (reference 15).

Lines 228-237 - In this paragraph the authors don’t explain the differences between the three markets analyzed to try to justify the differences of contamination rates.

Lines 240-241 – “Banana was the least contaminated fruit (14.9%).”

Line 248 – “A. lumbricoides”.

Line 249 – “was” instead of “were”.

Line 258 and 260 – “S. stercoralis”.

Line 261 – “Strongyloides spp.”.

Line 264 – In the begin of a paragrapgh don’t use an abbreviation of a name “Entamoeba histolytica/dispar cysts”.

Line 266 – Authors refer to two articles “references 10 and 13” but don’t explain what was the deference of findings.

Line 270 – Beginning of paragraph “Hymenolepis nana”.

Lines 275, 278, 283, 285 – “Cryptosporidium”.

Lines 275-279 - the authors compare the results they obtained with the results of other studies, but do not indicate what these other results were.

Line 286, 289 – “Toxocara spp.”

Line 291 – “Toxocara eggs”

Line 293 – “Fasciola”

Lines 295-296 – Authors write is not a “Some Fasciola spp are known to be neglected zoonotic diseases [25]”. Fasciola is a parasite that causes disease. Consider revision the phrase.

Lines 297-304 – In this area, the results that you obtained have to be compared with results of other studies, namely the article Salamandane et al, 2021.

6. PLOS authors have the option to publish the peer review history of their article (what does this mean?). If published, this will include your full peer review and any attached files.

Reviewer #1: **Yes: **Maria Cristina Angelici

Reviewer #2: No

---

## [Author Response · Author response to Decision Letter 0]

5 May 2023

Here is a point-by-point response to the reviewers’ comments and concerns;

Reviewer #1: 

“This study has been well carried out and shows how much dirty vegetables and fruits could transmit parasitic infections to humans. The experimentally procedure and statistically analysis is done with classical methods and the English grammatical form is corrected and well understandable. The only note to Authors is: how do you can distinguish Giardia cysts at specific level with only the microscopical assay? It could be better to indicate this parasite presence with Giardia sp. definition. Please, note also that Cryptosporidium is incorrectly written (Cyreptosporidium) and that spp. does not require capital letter (Spp), in both Figure 2 and 3.”

We would like to thank the reviewer#1 for his constructive feedback. The manuscript was improved with the requested recommendations and each comment has been addressed as following:

Question 1: How do you can distinguish Giardia cysts at specific level with only the microscopical assay? It could be better to indicate this parasite presence with Giardia spp. definition.

Response: Modified as suggested by the reviewer in the revised version of the manuscript. Thank you!

Comment 2: Please, note also that Cryptosporidium is incorrectly written (Cyreptosporidium) and that spp. does not require capital letter (Spp), in both Figure 2 and 3.

Response: We fully agree with this comment and to address it, the following change was done: 

- “Cyreptosporidium” replaced with “Cryptosporidium”

- “Spp” replaced with “spp.” 

Reviewer #2: 

We would like to thank Reviewer #2 for his comments which contributed to improve the manuscript. The manuscript was improved with the requested recommendations and each comment has been addressed. Additional details and discussion was added to clarify and compare our data. Each of the comments was addressed as following: 

Major comments: 

Comment 1: All rounding to decimal places must be checked as there are errors in all tables.

Response: All rounding to decimal places are thoroughly checked and errors are corrected in the revised version of the manuscript. Thank you!

Comment 2: The abbreviation “spp.” must always be written in lower case and not italics. Do the respective changes in all text and tables.

Response: Corrected throughout the entire text and tables in the revised version of the manuscript. Thank you!

Comment 3: Authors should always consider using the same designation for Giardia, lamblia or intestinalis (only one name), throughout the entire text.

Response: This comment was also done by Reviewer 1 and to address it, the designations for “Giardia, lamblia or intestinalis” throughout the entire text are modified with “Giardia spp.” by considering the difficulty microscopical assay for identification of the detected Giardia cysts in this study.

Comment 4: The authors do not explain whether the collection of samples was made in the rainy season or in the dry season, or summer or winter. The fact that they say that the collection was made from July to September does not indicate, for the reader, the degree of rainfall and temperature.

Response: We fully agree with this comment and to address it, the sample collection season is added as “during major rainy season” (Line 117). Degree of rain fall and temperature is also included (Lines 121-122).

Comment 5: In the introduction, results and discussion, the authors did not take into account a similar study that was carried out in Mozambique and published in 2021, involving similar populations. This study should be introduced in this article and compared with the results you obtained because it is a recent study, carried out in East Africa.

“Salamandane et al.,. (2021). Occurrence of intestinal parasites of public health significance in fresh horticultural products sold in Maputo markets and supermarkets, Mozambique. Microorganisms, 9, 1806. https://doi.org/10.3390/microorganisms9091806”

Response: To address this comment the following information from requested references was added in the introduction and discussion sections: 

“There are plenty of open markets where fruits and vegetables are sold in Africa. Fresh produce is exposed to the environment in these markets, including domestic animals, which add to the risk of food contamination [6, 9, 10, 11]. On this continent, studies on the prevalence of intestinal parasites associated with vegetables and fruits have been performed mostly in Mozambique, and Ghana [9, 10, 11].”(Lines 97-101)

Minor comments: 

Background:

Line 105 – “vegetables and fruits….”

Response: Corrected (Line 110) 

Parasitological examination: 

Line 145, 147 – “modified Ziehl–Neelsen….”

Response: Corrected (Line 150, 152) 

Line 145 – Since Cryptosporidium is not a coccidia, authors should write “detection of coccidian and Cryptosporidium oocysts….”

Response: Corrected (Line 150-151) 

Results:

Line 162 – “37.0%” instead of “37.3%”

Response: Corrected; “37.3%” replaced with “37 %”( Line 176). Thank you!

Line 172 – “ Cryptosporidium spp.”

Response: Corrected (Line 185) 

Lines 173-174 – “S. stercoralis”

Response: Corrected (Line 186) 

Line 175 – “Cryptosporidium (9.2%) oocysts”

Response: Corrected (Line 188) 

Line 175 – which forms of development were identified for Toxocara and Fasciola?

Response: the identified parasitic stages for both Toxocara and Fasciola parasites were eggs. The detected forms of development was depicted in the revised version of the manuscript (Line188)

Lines 181-182 – “detected in carrot 63.8% (30), followed by lettuce 63.1% (29) and Cabbage 61.7% (29). Banana with a prevalence of 14.9% (7)…”

Response: Corrected (Line 193-194)

Line 183 – “A. lumbricodes was abundantly…”

Response: Corrected (Line 195) 

Discussion: 

Lines 211-214 – The names of vegetables and fruits should be in low case “lettuce….”

Response: Corrected (Line 223-226)

Line 221 – Authors should discuss the difference of contamination rate found in their study and in the study done also in Kenya (reference 15).

Response: To address this comment the following information was added: 

“Our finding is explained by the fact that such markets have more frequent running water during the main rainy season, which increases the likelihood that producers and vendors will wash their products thoroughly. Another hypothesis is assuming that parasite forms on the surface of vegetables are washed away by rain [9]. Moreover, since 2015, water, sanitation, and hygiene (WASH) activities, health education, and biannual deworming programs for helminthes have been conducted and substantially decreased environmental fecal contamination as well as parasitic contamination in Ethiopia [6]. The difference may also be due to variations in the types and quantities of fruits and vegetables evaluated.” (Lines 231-240)

Lines 228-237 - In this paragraph the authors don’t explain the differences between the three markets analyzed to try to justify the differences of contamination rates.

Response: the differences of contamination rates among the three markets analyzed are depicted as “The differences between the three markets might be due to the different sources of vegetables as well as the hygienic practices in handling and washing by different sellers. Moreover, the high prevalence of intestinal parasites on vegetables is due to the fact that these open-air markets were characterized by poor sewage disposal systems, and vegetables were brought in from rural areas around these markets.” (246-250) 

Lines 240-241 – “Banana was the least contaminated fruit (14.9%).”

Response: Corrected (Line 253)

Line 248 – “A. lumbricoides”.

Response: Corrected (Line 263)

Line 249 – “was” instead of “were”.

Response: Corrected (Line 263)

Line 258 and 260 – “S. stercoralis”.

Response: Corrected (Line 271 & 273)

Line 261 – “Strongyloides spp.”

Response: Corrected (Line 274)

Line 264 – In the begin of a paragrapgh don’t use an abbreviation of a name “Entamoeba histolytica/dispar cysts”.

Response: Corrected (Line 277)

Line 266 – Authors refer to two articles “references 10 and 13” but don’t explain what was the deference of findings.

Response: the paragraph rephrased as the following “However, the rate of contamination was lower as compared to a study finding from Arba Minch, Ethiopia [13]. Variation in the prevalence of E. histolytica/E. dispar and Giardia spp. might be attributed to the long periods of survival of the cysts under cool and moist conditions and variations in geographical distribution [6].” (Line 279-280)

Line 270 – Beginning of paragraph “Hymenolepis nana”.

Response: Corrected (Line 283)

Lines 275, 278, 283, 285 – “Cryptosporidium”.

Response: Corrected (Line 288, 296, 301, 304)

Lines 275-279 - the authors compare the results they obtained with the results of other studies, but do not indicate what these other results were.

Response: Corrected (Line 288-296)

Line 286, 289 – “Toxocara spp.”

Response: Corrected (Lines 305, 308)

Line 291 – “Toxocara eggs”

Response: Corrected (Line 310)

Line 293 – “Fasciola”

Response: Corrected (Line 312)

Lines 295-296 – Authors write is not a “Some Fasciola spp are known to be neglected zoonotic diseases [25]”. Fasciola is a parasite that causes disease. Consider revision the phrase.

Response: We rephrased the sentence as requested by the reviewer (Line 314-315)

Lines 297-304 – In this area, the results that you obtained have to be compared with results of other studies, namely the article Salamandane et al, 2021.

Response: Corrected (Line 323-327)

---

## [Decision Letter · Decision Letter 1]

25 Jul 2023

PONE-D-23-01852R1Parasitic contamination of fresh vegetables and fruits sold in open-air markets in peri-urban areas of Jimma City, Oromia, Ethiopia: A community-based cross-sectional studyPLOS ONE

Dear Dr. Abamecha,

Thank you for submitting your manuscript to PLOS ONE. After careful consideration, we feel that it has merit but does not fully meet PLOS ONE’s publication criteria as it currently stands. Therefore, we invite you to submit a revised version of the manuscript that addresses the points raised during the review process.

We look forward to receiving your revised manuscript.

Kind regards,

Adriana Calderaro

Academic Editor

PLOS ONE

Journal Requirements:

Reviewers' comments:

Reviewer's Responses to Questions

**Comments to the Author**

1. If the authors have adequately addressed your comments raised in a previous round of review and you feel that this manuscript is now acceptable for publication, you may indicate that here to bypass the “Comments to the Author” section, enter your conflict of interest statement in the “Confidential to Editor” section, and submit your "Accept" recommendation.

Reviewer #1: All comments have been addressed

Reviewer #2: (No Response)

2. Is the manuscript technically sound, and do the data support the conclusions?

Reviewer #1: (No Response)

Reviewer #2: Yes

3. Has the statistical analysis been performed appropriately and rigorously? 

Reviewer #1: (No Response)

Reviewer #2: Yes

4. Have the authors made all data underlying the findings in their manuscript fully available?

Reviewer #1: (No Response)

Reviewer #2: Yes

5. Is the manuscript presented in an intelligible fashion and written in standard English?

Reviewer #1: (No Response)

Reviewer #2: Yes

6. Review Comments to the Author

Reviewer #1: (No Response)

Reviewer #2: Comments:

Some decimal places are incorrect in table 1, consider reviewing one by one:

2nd line - Lettuce, 3rd column the 5 should be 63.0;

In the results for Hora Gibe Market, the % for avocado are incorrect 8/15=53.3%;

The same for the total % 13/51=25.5%

The same in the results of Jiren Market

In Table 2 there are also mistakes with two of the decimals.

The data in table 2 are not uniformly presented. The first three lines describe the parasites identified and the forms found, but the same does not happen for Giardia spp., H. nana, Cryptosporidium spp., Toxocara spp. and Fasciola spp.

In Line 315 – authors should write, “Fascioliasis is known to be a neglected zoonosis”

7. PLOS authors have the option to publish the peer review history of their article (what does this mean?). If published, this will include your full peer review and any attached files.

Reviewer #1: No

Reviewer #2: **Yes: **Olga Matos

---

## [Author Response · Author response to Decision Letter 1]

28 Jul 2023

Here is a point-by-point response to the reviewers’ comments and concerns;

Reviewer #2: Comments:

Some decimal places are incorrect in table 1, consider reviewing one by one:

2nd line - Lettuce, 3rd column the 5 should be 63.0;

In the results for Hora Gibe Market, the % for avocado are incorrect 8/15=53.3%;

The same for the total % 13/51=25.5%

The same in the results of Jiren Market

Response: The decimal places are thoroughly checked again and errors are corrected in the revised version of the manuscript. Thank you!!

In Table 2 there are also mistakes with two of the decimals.

Response: Corrected 

The data in table 2 are not uniformly presented. The first three lines describe the parasites identified and the forms found, but the same does not happen for Giardia spp., H. nana, Cryptosporidium spp., Toxocara spp. and Fasciola spp.

Response: We fully agree with this comment and to keep the uniformity, only the parasites identified are presented in the revised version of the manuscript. 

In Line 315 – authors should write, “Fascioliasis is known to be a neglected zoonosis”

Response: Corrected (Line 315). Thank you!!

---

## [Editor Report · Decision Letter 2]

21 Aug 2023

Dear Dr. Abamecha,

Thank you for submitting your manuscript to PLOS ONE, and for responding to our recent requests regarding your submission. Unfortunately, in our final editorial checks of the documents that you supplied, we have concluded that your submission does not meet our ethical requirements for human subjects research submissions. We will therefore be overturning the provisional editorial accept decision, and will reject this manuscript.

PLOS ONE requires that research meets all applicable standards for the ethics of experimentation and research integrity (http://journals.plos.org/plosone/s/human-subjects-research). We reserve the right to reject any submission that does not meet our internal ethical standards, which in some cases are more stringent than local ethical standards.

In this case, we understand that you obtained ethical approval from your local board after you started this study. PLOS ONE does not accept retrospective ethical approval for research involving human participants.

As a result of this concern, we cannot consider the manuscript for publication. I am very sorry that this issue was identified at such a late stage.

Kind regards,

Emily Chenette

Editor in Chief

PLOS ONE

---

## [Author Response · Author response to Decision Letter 2]

8 Sep 2023

Date: August 24, 2023

Emily Chenette

Editor in Chief

PLOS ONE Journal

Issue: Appeal for Manuscript_ PONE-D-23-01852R2 - [EMID: dbc04e0f5d0ab20c]

This is in reference to the manuscript entitled “Parasitic contamination of fresh vegetables and fruits sold in open-air markets in peri-urban areas of Jimma City, Oromia, Ethiopia: A community-based cross-sectional study_ PONE-D-23-01852R2” and you informed us our Manuscript has been rejected due to the below reason:

 “We understand that you obtained ethical approval from your local board after you started this study. PLOS ONE does not accept retrospective ethical approval for research involving human participants.”

I would like to urge you to reconsider because of the ethical letter dated 4/08/2021 is a typographical error. I have attached an official letter from the Institutional review board (IRB) and the corrected version of the ethical clearance.

We hope this correction will make you aware of the right information. We apologize to you for the inconvenience it may cause you.

I look forward to hearing from you.

Kind regards,

Abdulhakim Abamecha , Ph.D.

Assistant Professor of Medical Microbiology & 

Tropical and Infectious Diseases

Head(Acting), School of Medical Laboratory Sciences,

Faculty of Health Sciences, Institute of Health,

Jimma University, Jimma, Ethiopia 

Tel.: +251 911 050437(Mobile) P.O.Box: 378 Jimma, Ethiopia

ORCID: https://orcid.org/0000-0003-4501-4759

 Alternative Email: abdulhakim.abamecha@ju.edu.et

 Website: https://ju.edu.et/health-institute/school-of-medical-laboratory-sciences/

---

## [Editor Report · Decision Letter 3]

30 Nov 2023

PONE-D-23-01852R3Parasitic contamination of fresh vegetables and fruits sold in open-air markets in peri-urban areas of Jimma City, Oromia, Ethiopia: A community-based cross-sectional studyPLOS ONE

Dear Dr. Abamecha,

Thank you for submitting your manuscript to PLOS ONE. After careful consideration, we feel that it has merit but does not fully meet PLOS ONE’s publication criteria as it currently stands. Therefore, we invite you to submit a revised version of the manuscript that addresses the points raised during the review process.

We look forward to receiving your revised manuscript.

Kind regards,

Jianhong Zhou

Staff Editor

PLOS ONE

Request from Staff Editors:

Before we can proceed, please provide a copy, for our internal records, of the study protocol approved by the Jimma University Institute of Health IRB REF No. IHRPGn/357/21

We thank for your attention to this query
---

## [Author Response · Author response to Decision Letter 3]

6 Dec 2023

Here is a point-by-point response to the staff Editors request;

Request from Staff Editors: Before we can proceed, please provide a copy, for our internal records, of the study protocol approved by the Jimma University Institute of Health IRB REF No. IHRPGn/357/21

Response: I have attached a copy of the study protocol that has been approved by IRB of Jimma University Institute of Health.

---

## [Editor Report · Decision Letter 4]

22 Jan 2024

Parasitic contamination of fresh vegetables and fruits sold in open-air markets in peri-urban areas of Jimma City, Oromia, Ethiopia: A community-based cross-sectional study

PONE-D-23-01852R4

Dear Dr. Abamecha,

We’re pleased to inform you that your manuscript has been judged scientifically suitable for publication and will be formally accepted for publication once it meets all outstanding technical requirements.

Kind regards,

Marcello Otake Sato, Ph.D., D.V.M.

Academic Editor

PLOS ONE
---

## [Editor Report · Acceptance letter]

1 Feb 2024

PONE-D-23-01852R4 

PLOS ONE

Dear Dr. Abamecha, 

I'm pleased to inform you that your manuscript has been deemed suitable for publication in PLOS ONE. Congratulations! Your manuscript is now being handed over to our production team.

Kind regards, 

on behalf of

Dr. Marcello Otake Sato 

Academic Editor

PLOS ONE